# Library Learning for Neurally-Guided Bayesian Program Induction

**Kevin Ellis**
MIT
ellisk@mit.edu

**Lucas Morales**
MIT
lucasem@mit.edu

**Mathias Sablé-Meyer**
ENS Paris-Saclay
mathsm@mit.edu

**Armando Solar-Lezama**
MIT
asolar@csail.mit.edu

**Joshua B. Tenenbaum**
MIT
jbt@mit.edu

## Abstract

Successful approaches to program induction require a hand-engineered domain-specific language (DSL), constraining the space of allowed programs and imparting prior knowledge of the domain. We contribute a program induction algorithm called $EC^2$ that learns a DSL while jointly training a neural network to efficiently search for programs in the learned DSL. We use our model to synthesize functions on lists, edit text, and solve symbolic regression problems, showing how the model learns a domain-specific library of program components for expressing solutions to problems in the domain.

## 1   Introduction

Much of everyday human thinking and learning can be understood in terms of program induction: constructing a procedure that maps inputs to desired outputs, based on observing example input-output pairs. People can induce programs flexibly across many different domains, and remarkably, often from just one or a few examples. For instance, if shown that a text-editing program should map "Jane Morris Goodall" to "J. M. Goodall", we can guess it maps "Richard Erskine Leakey" to "R. E. Leakey"; if instead the first input mapped to "Dr. Jane", "Goodall, Jane", or "Morris", we might have guessed the latter should map to "Dr. Richard", "Leakey, Richard", or "Erskine", respectively.

The FlashFill system [1] developed by Microsoft researchers and now embedded in Excel solves problems such as these and is probably the best known practical program-induction algorithm, but researchers in programming languages and AI have built successful program induction algorithms for many applications, such as handwriting recognition and generation [2], procedural graphics [3], cognitive modeling [4], question answering [5] and robot motion planning [6], to name just a few. These systems work in different ways, but most hinge upon having a carefully engineered **Domain Specific Language (DSL)**. This is especially true for systems such as FlashFill that aim to induce a wide range of programs very quickly, in a few seconds or less. DSLs constrain the search over programs with strong prior knowledge in the form of a restricted set of programming primitives tuned to the needs of the domain: for text editing, these are operations like appending strings and splitting on characters.

In this work, we consider the problem of building agents that learn to solve program induction tasks, and also the problem of acquiring the prior knowledge necessary to quickly solve these tasks in a new domain. Representative problems in three domains are shown in Table 1. Our solution is an algorithm that grows or boostraps a DSL while jointly training a neural network to help write programs in the increasingly rich DSL.

| List Functions | Text Editing | Symbolic Regression |
|---|---|---|

**Programs & Tasks**

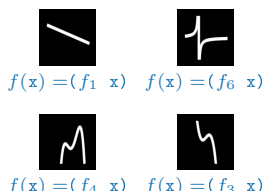

List Functions:
```
[7 2 3]→[7 3]
[1 2 3 4]→[3 4]
[4 3 2 1]→[4 3]
f(ℓ) = (f₀ ℓ (λ (x)
          (> x 2)))

[2 7 8 1]→8
[3 19 14]→19
f(ℓ) = (f₁ ℓ)

[7 3]→False
[3]→False
[9 0 0]→True
[0]→True
[0 7 3]→True
f(ℓ) = (f₂ ℓ 0)
```

Text Editing:
```
+106 769-438→106.769.438
+83 973-831→83.973.831
f(s) = (f₀ "." "-"
         (f₀ "." " "
           (cdr s)))

Temple Anna H →TAH
Lara Gregori→LG
f(s) = (f₂ s)
```

Symbolic Regression:
$$f(x) = (f_1\ x) \qquad f(x) = (f_6\ x)$$
$$f(x) = (f_4\ x) \qquad f(x) = (f_3\ x)$$

**DSL**

List Functions:
```
f₀(ℓ,p) = (foldr ℓ nil (λ (x a)
          (if (p x) (cons x a) a)))
(f₀: Higher-order filter function)
f₁(ℓ) = (foldr ℓ 0 (λ (x a)
          (if (> a x) a x)))
(f₁: Maximum element in list ℓ)
f₂(ℓ,k) = (foldr ℓ (is-nil ℓ)
          (λ (x a) (if a a (= k x))))
(f₂: Whether ℓ contains k)
```

Text Editing:
```
f₀(s,a,b) = (map (λ (x)
          (if (= x a) b x)) s)
(f₀: Performs character substitution)
f₁(s,c) = (foldr s s (λ (x a)
          (cdr (if (= c x) s a))))
(f₁: Drop characters from s until c reached)
f₂(s) = (unfold s is-nil car
          (λ (z) (f₁ z " ")))
(f₂: Abbreviates a sequence of words)
f₃(a,b) = (foldr a b cons)
(f₃: Concatenate strings a and b)
```

Symbolic Regression:
```
f₀(x) = (+ x real)
f₁(x) = (f₀ (* real x))
f₂(x) = (f₁ (* x (f₀ x)))
f₃(x) = (f₀ (* x (f₂ x)))
f₄(x) = (f₀ (* x (f₃ x)))
(f₄: 4th order polynomial)
f₅(x) = (/ real x)
f₆(x) = (f₄ (f₀ x))
(f₆: rational function)
```

Table 1: Top: Tasks from each domain, each followed by the programs $EC^2$ discovers for them. Bottom: Several examples from learned DSL. Notice that learned DSL primitives can call each other, and that $EC^2$ rediscovers higher-order functions like `filter` ($f_0$ in List Functions)

Because any computable learning problem can in principle be cast as program induction, it is important to delimit our focus. In contrast to computer assisted programming [7] or genetic programming [8], our goal is not to automate software engineering, to learn to synthesize large bodies of code, or to learn complex programs starting from scratch. Ours is a basic AI goal: capturing the human ability to learn to think flexibly and efficiently in new domains — to learn what you need to know about a domain so you don't have to solve new problems starting from scratch. We are focused on the kinds of problems that humans can solve relatively quickly, once they acquire the relevant domain expertise. These correspond to tasks solved by short programs — if you have an expressive DSL. Even with a good DSL, program search may be intractable; so we amortize the cost of search by training a neural network to assist the search procedure.

Our algorithm takes inspiration from several ways that skilled human programmers have learned to code: skilled coders build libraries of reusable subroutines that are shared across related programming tasks, and can be composed to generate increasingly complex and powerful subroutines. In text editing, a good library should support routines for splitting on characters, but also specialize these routines to split on particular characters such as spaces or commas that are frequently used to delimit substrings across tasks. Skilled coders also learn to recognize what kinds of programming idioms and library routines would be useful for solving the task at hand, even if they cannot instantly work out the details. In text editing, one might learn that if outputs are consistently shorter than inputs, removing characters is likely to be part of the solution; if every output contains a constant substring (e.g., "Dr."), inserting or appending that constant string is likely to be a subroutine.

Our $EC^2$ (ECC, for **E**xplore/**C**ompress/**C**ompile) algorithm incorporates these insights by iterating through three steps. The **Explore** step takes a given set of **tasks**, typically several hundred, and explores the space of programs, searching for compact programs that solve these tasks, guided by the current DSL and neural network. The **Compress** step grows the library (or DSL) of domain-specific subroutines which allow the agent to more compactly write programs in the domain; it modifies the structure of the DSL by discovering regularities across programs found during the Explore step, compressing them to distill out common code fragments across successful programs. The **Compile** step improves the search procedure by training a neural network to write programs in the current DSL, in the spirit of "amortized" or "compiled" inference [9, 10].

The learned DSL effectively encodes a prior on programs likely to solve tasks in the domain, while the neural net looks at the example input-output pairs for a specific task and produces a "posterior" for programs likely to solve that specific task. The neural network thus functions as a **recognition model** supporting a form of approximate Bayesian program induction, jointly trained with a **generative model** for programs encoded in the DSL, in the spirit of the Helmholtz machine [11]). The recognition

model ensures that searching for programs remains tractable even as the DSL (and hence the search space for programs) expands.

We apply $EC^2$ to three domains: list processing; text editing (in the style of FlashFill [1]); and symbolic regression. For each of these we initially provide a generic set of programming primitives. Our algorithm then constructs its own DSL for expressing solutions in the domain (Tbl. 1).

Prior work on program learning has largely assumed a fixed, hand-engineered DSL, both in classic symbolic program learning approaches (e.g., Metagol: [12], FlashFill: [1]), neural approaches (e.g., RobustFill: [13]), and hybrids of neural and symbolic methods (e.g., Neural-guided deductive search: [14], DeepCoder: [15]). A notable exception is the EC algorithm [16], which also learns a library of subroutines. We find EC motivating, and go beyond it and other prior work through the following contributions: (1) We show how to learn-to-learn programs in an expressive Lisp-like programming language, including conditionals, variables, and higher-order recursive functions; (2) We give an algorithm for learning DSLs, built on a formalism known as Fragment Grammars [17]; and (3) We give a hierarchical Bayesian framing of the problem that allows joint inference of the DSL and neural recognition model.

## 2 The $EC^2$ Algorithm

We first mathematically describe our 3-step algorithm as an inference procedure for a hierarchical Bayesian model (Section 2.1), and then describe each step algorithmically in detail (Section 2.2-2.4).

### 2.1 Hierarchical Bayesian Framing

$EC^2$ takes as input a set of *tasks*, written $X$, each of which is a program synthesis problem. It has at its disposal a domain-specific *likelihood model*, written $\mathbb{P}[x|p]$, which scores the likelihood of a task $x \in X$ given a program $p$. Its goal is to solve each of the tasks by writing a program, and also to infer a DSL, written $\mathcal{D}$. We equip $\mathcal{D}$ with a real-valued weight vector $\theta$, and together $(\mathcal{D}, \theta)$ define a generative model over programs. We frame our goal as maximum a posteriori (MAP) inference of $(\mathcal{D}, \theta)$ given $X$. Writing $J$ for the joint probability of $(\mathcal{D}, \theta)$ and $X$, we want the $\mathcal{D}^*$ and $\theta^*$ solving:

$$J(\mathcal{D}, \theta) \triangleq \mathbb{P}[\mathcal{D}, \theta] \prod_{x \in X} \sum_p \mathbb{P}[x|p]\mathbb{P}[p|\mathcal{D}, \theta]$$

$$\mathcal{D}^* = \arg\max_{\mathcal{D}} \int J(\mathcal{D}, \theta) \, \mathrm{d}\theta \qquad \theta^* = \arg\max_{\theta} J(\mathcal{D}^*, \theta) \tag{1}$$

The above equations summarize the problem from the point of view of an ideal Bayesian learner. However, Eq. 1 is wildly intractable because evaluating $J(\mathcal{D}, \theta)$ involves summing over the infinite set of all programs. In practice we will only ever be able to sum over a finite set of programs. So, for each task, we define a finite set of programs, called a *frontier*, and only marginalize over the frontiers:

**Definition.** A *frontier of task* $x$, written $\mathcal{F}_x$, is a finite set of programs s.t. $\mathbb{P}[x|p] > 0$ for all $p \in \mathcal{F}_x$.

Using the frontiers we define the following intuitive lower bound on the joint probability, called $\mathscr{L}$:

$$J \geq \mathscr{L} \triangleq \mathbb{P}[\mathcal{D}, \theta] \prod_{x \in X} \sum_{p \in \mathcal{F}_x} \mathbb{P}[x|p]\mathbb{P}[p|\mathcal{D}, \theta] \tag{2}$$

$EC^2$ does approximate MAP inference by maximizing this lower bound on the joint probability, alternating maximization w.r.t. the frontiers (Explore) and the DSL (Compress):
**Explore: Maxing $\mathscr{L}$ w.r.t. the frontiers.** Here $(\mathcal{D}, \theta)$ is fixed and we want to find new programs to add to the frontiers so that $\mathscr{L}$ increases the most. $\mathscr{L}$ most increases by finding programs where $\mathbb{P}[x|p]\mathbb{P}[p|\mathcal{D}, \theta]$ is large.
**Compress: Maxing $\int \mathscr{L} \, \mathrm{d}\theta$ w.r.t. the DSL.** Here $\{\mathcal{F}_x\}_{x \in X}$ is held fixed, and so we can evaluate $\mathscr{L}$. Now the problem is that of searching the discrete space of DSLs and finding one maximizing $\int \mathscr{L} \, \mathrm{d}\theta$. Once we have a DSL $\mathcal{D}$ we can update $\theta$ to $\arg\max_{\theta} \mathscr{L}(\mathcal{D}, \theta, \{\mathcal{F}_x\})$.

Searching for programs is hard because of the large combinatorial search space. We ease this difficulty by training a neural recognition model, $q(\cdot|\cdot)$, during the Compile step: $q$ is trained to approximate

the posterior over programs, $q(p|x) \approx \mathbb{P}[p|x, \mathcal{D}, \theta] \propto \mathbb{P}[x|p]\mathbb{P}[p|\mathcal{D}, \theta]$, thus amortizing the cost of finding programs with high posterior probability.

**Compile: learning to tractably maximize $\mathscr{L}$ w.r.t. the frontiers.** Here we train $q(p|x)$ to assign high probability to programs $p$ where $\mathbb{P}[x|p]\mathbb{P}[p|\mathcal{D}, \theta]$ is large, because including those programs in the frontiers will most increase $\mathscr{L}$. We train $q$ both on programs found during the Explore step and on samples from the current DSL.

Crucially, each of these three steps bootstraps the others (Fig. 1): improving either the DSL or the recognition model makes search easier, so we find more programs solving tasks; both improving the DSL and solving more tasks expands the training data for the recognition model; and finding more programs that solve tasks gives more data from which to learn a DSL.

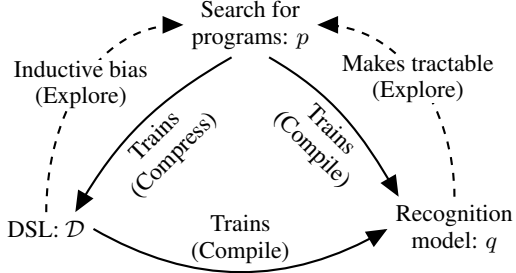

Figure 1: How these steps bootstrap each other.

## 2.2 Explore: Searching for Programs

Now our goal is to search for programs solving the tasks. We use the simple approach of enumerating programs from the DSL in decreasing order of their probability, and then checking if a program $p$ assigns positive probability to a task ($\mathbb{P}[x|p] > 0$); if so, we incorporate $p$ into the frontier $\mathcal{F}_x$.

To make this concrete we need to define what programs actually are and what form $\mathbb{P}[p|\mathcal{D}, \theta]$ takes. We represent programs as $\lambda$-calculus expressions. $\lambda$-calculus is a formalism for expressing functional programs that closely resembles Lisp, including variables, function application, and the ability to create new functions. Throughout this paper we will write $\lambda$-calculus expressions in Lisp syntax. Our programs are all strongly typed. We use the Hindley-Milner polymorphic typing system [18] which is used in functional programming languages like OCaml and Haskell. We now define DSLs:

**Definition:** $(\mathcal{D}, \theta)$. A DSL $\mathcal{D}$ is a set of typed $\lambda$-calculus expressions. A weight vector $\theta$ for a DSL $\mathcal{D}$ is a vector of $|\mathcal{D}| + 1$ real numbers: one number for each DSL element $e \in \mathcal{D}$, written $\theta_e$ and controlling the probability of $e$ occurring in a program, and a weight controlling the probability of a variable occurring in a program, $\theta_{\mathrm{var}}$.

Together with its weight vector, a DSL defines a distribution over programs, $\mathbb{P}[p|\mathcal{D}, \theta]$. In the supplement, we define this distribution by specifying a procedure for drawing samples from $\mathbb{P}[p|\mathcal{D}, \theta]$.

Why enumerate, when the program synthesis community has invented many sophisticated algorithms that search for programs? [7, 19, 20, 21, 22, 23]. We have two reasons: (1) A key point of our work is that learning the DSL, along with a neural recognition model, can make program induction tractable, even if the search algorithm is very simple. (2) Enumeration is a general approach that can be applied to any program induction problem. Many of these more sophisticated approaches require special conditions on the space of programs.

However, a drawback of enumerative search is that we have no efficient means of solving for arbitrary constants that might occur in a program. In Sec. 4, we will show how to find programs with real-valued constants by automatically differentiating through the program and setting the constants using gradient descent.

## 2.3 Compile: Learning a Neural Recognition Model

The purpose of training the recognition model is to amortize the cost of searching for programs. It does this by learning to predict, for each task, programs with high likelihood according to $\mathbb{P}[x|p]$ while also being probable under the prior $(\mathcal{D}, \theta)$. Concretely, the recognition model $q$ predicts, for each task $x \in X$, a weight vector $q(x) = \theta^{(x)} \in \mathbb{R}^{|\mathcal{D}|+1}$. Together with the DSL, this defines a distribution over programs, $\mathbb{P}[p|\mathcal{D}, \theta = q(x)]$. We abbreviate this distribution as $q(p|x)$. The crucial aspect of this framing is that the neural network leverages the structure of the learned DSL, so it is *not* responsible for generating programs wholesale. We share this aspect with DeepCoder [15] and [24].

How should we get the data to train $q$? This is non-obvious because we are considering a weakly supervised setting (i.e., learning only from tasks and not from task/program pairs). One approach is to sample programs from the DSL, run them to get their input/outputs, and then train $q$ to predict the program from the input/outputs. This is like how the wake-sleep algorithm for the Helmholtz machine trains its recognition model during its sleep phase [25]. The advantage of training on samples, or "fantasies," is that we can draw unlimited samples from the DSL, training on a large amount of data. Another approach is to train $q$ on the (program, task) pairs discovered by the Explore step. The advantage here is that the training data is much higher quality, because we are training on real tasks. Due to these complementary advantages, we train on both these sources of data.

Formally, $q$ should approximate the true posteriors over programs: minimizing the expected KL-divergence, $\mathbb{E}\left[\text{KL}\left(\mathbb{P}[p|x, \mathcal{D}, \theta]\|q(p|x)\right)\right]$, equivalently maximizing $\mathbb{E}[\sum_p \mathbb{P}[p|x, \mathcal{D}, \theta] \log q(p|x)]$, where the expectation is taken over tasks. Taking this expectation over the empirical distribution of tasks trains $q$ on the real data; taking it over samples from the generative model trains $q$ on "fantasies." The objective for a recognition model ($\mathcal{L}_{\text{RM}}$) combines the fantasy ($\mathcal{L}_{\text{f}}$) and real-data ($\mathcal{L}_{\text{r}}$) objectives, $\mathcal{L}_{\text{RM}} = \mathcal{L}_{\text{r}} + \mathcal{L}_{\text{f}}$:

$$\mathcal{L}_{\text{f}} = \mathbb{E}_{(p,x)\sim(\mathcal{D},\theta)}\left[\log q(p|x)\right] \quad \mathcal{L}_{\text{r}} = \mathbb{E}_{x\sim X}\left[\sum_{p\in\mathcal{F}_x} \frac{\mathbb{P}\left[x, p|\mathcal{D}, \theta\right]}{\sum_{p'\in\mathcal{F}_x}\mathbb{P}\left[x, p'|\mathcal{D}, \theta\right]} \log q(p|x)\right]$$

### 2.4 Compress: Learning a Generative Model (a DSL)

The purpose of the DSL is to offer a set of abstractions that allow an agent to easily express solutions to the tasks at hand. Intuitively, we want the algorithm to look at the frontiers and generalize beyond them, both so the DSL can better express the current solutions, and also so that the DSL might expose new abstractions which will later be used to discover more programs. Formally, we want the DSL maximizing $\int \mathscr{L} \, d\theta$ (Sec. 2.1). We replace this marginal with an AIC approximation, giving the following objective for DSL induction:

$$\log \mathbb{P}[\mathcal{D}] + \arg\max_\theta \sum_{x\in X} \log \sum_{p\in\mathcal{F}_x} \mathbb{P}[x|p]\mathbb{P}[p|\mathcal{D}, \theta] + \log \mathbb{P}[\theta|\mathcal{D}] - \|\theta\|_0 \tag{3}$$

We induce a DSL by searching locally through the space of DSLs, proposing small changes to $\mathcal{D}$ until Eq. 3 fails to increase. The search moves work by introducing new $\lambda$-expressions into the DSL. We propose these new expressions by extracting fragments of programs already in the frontiers (Tbl. 2). An important point here is that we are *not* simply adding subexpressions of programs to $\mathcal{D}$, as done in the EC algorithm [16] and other prior work [26]. Instead, we are extracting fragments that unify with programs in the frontiers. This idea of storing and reusing fragments of expressions comes from Fragment Grammars [17] and Tree-Substitution Grammars [27], and is closely related to the idea of antiunification [28, 29]. Care must be taken to ensure that this 'fragmenting' obeys variable scoping rules; Section 4 of the supplement gives an overview of Fragment Grammars and how we adapt them to the lexical scoping rules of $\lambda$-calculus. To define the prior distribution over $(\mathcal{D}, \theta)$, we penalize the syntactic complexity of the $\lambda$-calculus expressions in the DSL, defining $\mathbb{P}[\mathcal{D}] \propto \exp(-\lambda \sum_{p\in\mathcal{D}} \text{size}(p))$ where $\text{size}(p)$ measures the size of the syntax tree of program $p$, and place a symmetric Dirichlet prior over the weight vector $\theta$.

Putting all these ingredients together, Alg. 1 describes how we combine program search, recognition model training, and DSL induction. For added robustness, we interleave an extra program search step (Explore) before training the recognition model, and just enumerate from the prior $(\mathcal{D}, \theta)$ during this extra Explore step.

## 3 Programs that manipulate sequences

We apply $\text{EC}^2$ to list processing (Section 3.1) and text editing (Section 3.2). For both these domains we use a bidirectional GRU [30] for the recognition model, and initially provide the system with a generic set of list processing primitives: `foldr`, `unfold`, `if`, `map`, `length`, `index`, `=`, `+`, `-`, `0`, `1`, `cons`, `car`, `cdr`, `nil`, and `is-nil`.

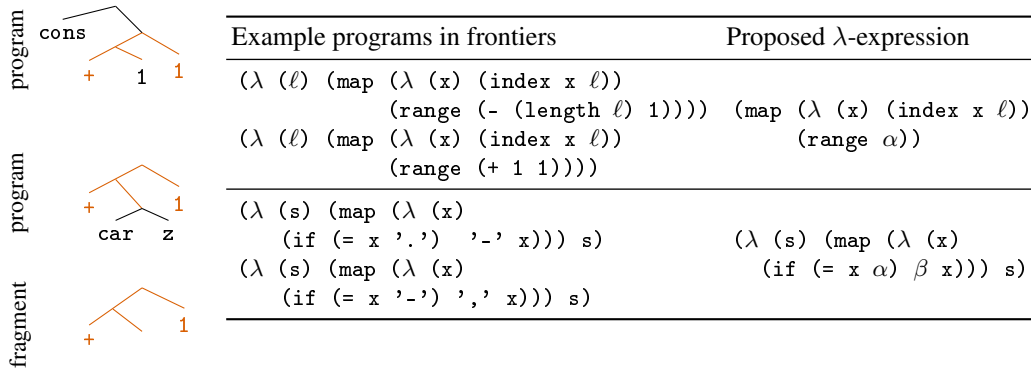

Figure 2: **Left:** syntax trees of two programs sharing common structure, highlighted in orange, from which we extract a fragment and add it to the DSL (bottom). **Right:** actual programs, from which we extract fragments that (top) slice from the beginning of a list or (bottom) perform character substitutions.

---

**Algorithm 1** The EC$^2$ Algorithm

**Input:** Initial DSL $\mathcal{D}$, set of tasks $X$, iterations $I$
**Hyperparameters:** Enumeration timeout $T$
Initialize $\theta \leftarrow$ uniform
**for** $i = 1$ **to** $I$ **do**
    For each task $x \in X$, set $\mathcal{F}_x^\theta \leftarrow \{p|p \in \mathrm{enum}(\mathcal{D}, \theta, T) \text{ if } \mathbb{P}[x|p] > 0\}$     **(Explore)**
    $q \leftarrow$ train recognition model, maximizing $\mathcal{L}_{\mathrm{RM}}$ (see Sec. 2.3)     **(Compile)**
    For each task $x \in X$, set $\mathcal{F}_x^q \leftarrow \{p|p \in \mathrm{enum}(\mathcal{D}, q(x), T) \text{ if } \mathbb{P}[x|p] > 0\}$  **(Explore)**
    $\mathcal{D}, \theta \leftarrow \mathrm{induceDSL}(\{\mathcal{F}_x^\theta \cup \mathcal{F}_x^q\}_{x \in X})$ (see Sec. 2.4)     **(Compress)**
**end for**
**return** $\mathcal{D}, \theta, q$

---

### 3.1 List Processing

Synthesizing programs that manipulate data structures is a widely studied problem in the programming languages community [20]. We consider this problem within the context of learning functions that manipulate lists, and which also perform arithmetic operations upon lists of numbers.

We created 236 human-interpretable list manipulation tasks, each with 15 input/output examples (Tbl. 2). Our data set is interesting in three major ways: many of the tasks require complex solutions; the tasks were not generated from some latent DSL; and the agent must learn to solve these complicated problems from only 236 tasks. Our data set assumes arithmetic operations as well as sequence operations, so we additionally provide our system with the following arithmetic primitives: `mod`, `*`, `>`, `is-square`, `is-prime`.

| Name | Input | Output |
|---|---|---|
| repeat-2 | [7 0] | [7 0 7 0] |
| drop-3 | [0 3 8 6 4] | [6 4] |
| rotate-2 | [8 14 1 9] | [1 9 8 14] |
| count-head-in-tail | [1 2 1 1 3] | 2 |
| keep-mod-5 | [5 9 14 6 3 0] | [5 0] |
| product | [7 1 6 2] | 84 |

Table 2: Some tasks in our list function domain. See the supplement for the complete data set.

We evaluated EC$^2$ on random 50/50 test/train split. Interestingly, we found that the recognition model provided little benefit for the training tasks. However, it yielded faster search times on held out tasks, allowing more tasks to be solved before timing out. The system composed 38 new subroutines, yielding a more expressive DSL more closely matching the domain (left of Tbl. 1, right of Fig. 2). See the supplement for a complete list of DSL primitives discovered by EC$^2$.

## 3.2   Text Editing

Synthesizing programs that edit text is a classic problem in the programming languages and AI literatures [24, 31], and algorithms that learn text editing programs ship in Microsoft Excel [1]. This prior work presumes a hand-engineered DSL. We show EC$^2$ can instead start out with generic sequence manipulation primitives and recover many of the higher-level building blocks that have made these other text editing systems successful.

Because our enumerative search procedure cannot generate string constants, we instead enumerate programs with string-valued parameters. For example, to learn a program that prepends "Dr.", we enumerate ($f_3$ string s) – where $f_3$ is the learned appending primitive (Fig. 1) — and then define $\mathbb{P}[x|p]$ by approximately marginalizing out the string parameters via a simple dynamic program. In Sec. 4, we will use a similar trick to synthesize programs containing real numbers, but using gradient descent instead of dynamic programming.

We trained our system on a corpus of 109 automatically generated text editing tasks, with 4 input/output examples each. After three iterations, it assembles a DSL containing a dozen new functions (center of Fig. 1) that let it solve all of the training tasks. But, how well does the learned DSL generalized to real text-editing scenarios? We tested, but did not train, on the 108 text editing problems from the SyGuS [32] program synthesis competition. Before any learning, EC$^2$ solves 3.7% of the problems with an average search time of 235 seconds. After learning, it solves 74.1%, and does so much faster, solving them in an average of 29 seconds. As of the 2017 SyGuS competition, the best-performing algorithm solves 82.4% of the problems. But, SyGuS comes with a different hand-engineered DSL *for each text editing problem.*[1] Here we learned a single DSL that applied generically to all of the tasks, and perform comparably to the best prior work.

## 4   Symbolic Regression: Programs from visual input

We apply EC$^2$ to symbolic regression problems. Here, the agent observes points along the curve of a function, and must write a program that fits those points. We initially equip our learner with addition, multiplication, and division, and task it with solving 100 symbolic regression problems, each either a polynomial of degree 1–4 or a rational function. The recognition model is a convolutional network that observes an image of the target function's graph (Fig. 3) — visually, different kinds of polynomials and rational functions produce different kinds of graphs, and so the recognition model can learn to look at a graph and predict what kind of function best explains it. A key difficulty, however, is that these problems are best solved with programs containing real numbers. Our solution to this difficulty is to enumerate programs with real-valued parameters, and then fit those parameters by automatically differentiating through the programs the system writes and use gradient descent to fit the parameters. We define the likelihood model, $\mathbb{P}[x|p]$, by assuming a Gaussian noise model for the input/output examples, and penalize the use of real-valued parameters using the BIC [33].

EC$^2$ learns a DSL containing 13 new functions, most of which are templates for polynomials of different orders or ratios of polynomials. It also learns to find programs that minimize the number of continuous degrees of freedom. For example, it learns to represent linear functions with the program `(* real (+ x real))`, which has two continuous degrees of freedom, and represents quartic functions using the invented DSL primitive $f_4$ in the rightmost column of Fig. 1 which has five continuous parameters. This phenomenon arises from our Bayesian framing — both the implicit bias towards shorter programs and the likelihood model's BIC penalty.

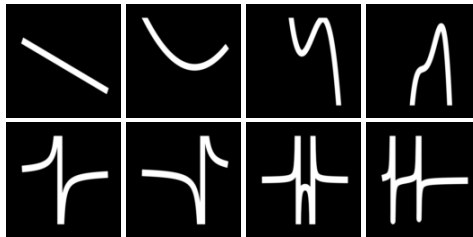

Figure 3: Recognition model input for symbolic regression. DSL learns subroutines for polynomials (top row) and rational functions (bottom row) while the recognition model jointly learns to look at a graph of the function (above) and predict which of those subroutines best explains the observation.

# 5 Quantitative Results

We compare with ablations of our model on held out tasks. The purpose of this ablation study is both to examine the role of each component of $EC^2$, as well as to compare with prior approaches in the literature: a head-to-head comparison of program synthesizers is complicated by the fact that each system, including ours, makes idiosyncratic assumptions about the space of programs and the statement of tasks. Nevertheless, much prior work can be modeled within our setup. We compare with the following ablations (Tbl 3; Fig 4):

**No NN:** lesions the recognition model.

**NPS**, which does not learn the DSL, instead learning the recognition model from samples drawn from the fixed DSL. We call this NPS (Neural Program Synthesis) because this is closest to how RobustFill [13] and DeepCoder [15] are trained.

**SE**, which lesions the recognition model and restricts the DSL learning algorithm to only add **S**ub**E**xpressions of programs in the frontiers to the DSL. This is how most prior approaches have learned libraries of functions [16, 34, 26].

**PCFG**, which lesions the recognition model and does not learn the DSL, but instead learns the parameters of the DSL ($\theta$), learning the parameters of a PCFG while not learning any of the structure.

**Enum**, which enumerates a frontier without any learning — equivalently, our first Explore step.

We are interested both in how many tasks the agent can solve and how quickly it can find those solutions. Tbl. 3 compares our model against these alternatives. We consistently improve on the baselines, and also find that lesioning the recognition model impairs the convergence of the algorithm, causing it to hit a lower 'plateau' after which it stops solving new tasks, following an initial spurt of learning (Fig. 4) – without the neural network, search becomes intractable. This lowered 'plateau' supports a view of the recognition model as a way of amortizing the cost of search.

|  | Ours | No NN | SE | NPS | PCFG | Enum |
|---|---|---|---|---|---|---|
| *List Processing* | | | | | | |
| % solved | **94%** | 79% | 71% | 35% | 62% | 37% |
| Solve time | 88s | 39s | 11s | 35s | 44s | 20s |
| *Text Editing* | | | | | | |
| % solved | **74%** | 43% | 30% | 33% | 0% | 4% |
| Solve time | 29s | 49s | 38s | 80s | – | 235s |
| *Symbolic Regression* | | | | | | |
| % solved | **84%** | 75% | 62% | 38% | 38% | 37% |
| Solve time | 24s | 40s | 28s | 31s | 55s | 29s |

Table 3: % held-out test tasks solved. Solve time: averaged over solved tasks.

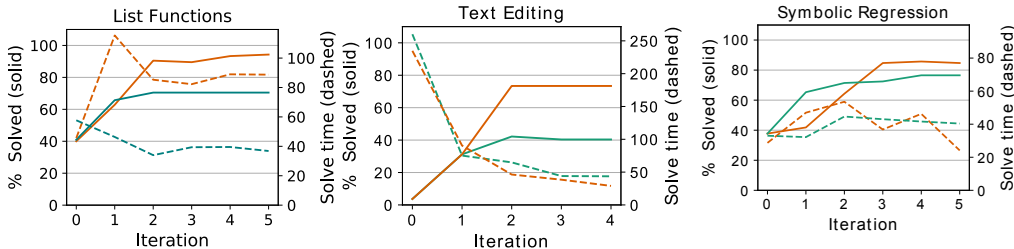

Figure 4: Learning curves for $EC^2$ both with (in orange) and without (in teal) the recognition model. Solid lines: % holdout testing tasks solved. Dashed lines: Average solve time.

# 6 Related Work

Our work is far from the first for learning to learn programs, an idea that goes back to Solomonoff [35]:

**Deep learning:** Much recent work in the ML community has focused on creating neural networks that regress from input/output examples to programs [13, 6, 24, 15]. $EC^2$'s recognition model draws heavily from this line of work, particularly from [24]. We see these prior works as operating in a different regime: typically, they train with strong supervision (i.e., with annotated ground-truth programs) on massive data sets (i.e., hundreds of millions [13]). Our work considers a weakly-supervised regime where ground truth programs are not provided and the agent must learn from at most a few hundred tasks, which is facilitated by our "Helmholtz machine" style recognition model.

**Inventing new subroutines for program induction:** Several program induction algorithms, most prominently the EC algorithm [16], take as their goal to learn new, reusable subroutines that are shared in a multitask setting. We find this work inspiring and motivating, and extend it along two dimensions: (1) we propose a new algorithm for inducing reusable subroutines, based on Fragment Grammars [17]; and (2) we show how to combine these techniques with bottom-up neural recognition models. Other instances of this related idea are [34], Schmidhuber's OOPS model [36], MagicHaskeller [37], Bayesian program merging [29], and predicate invention in Inductive Logic Programming [26]. Closely allied ideas have been applied to mining 'code idioms' from programs [38], and, concurrent with this work, using those idioms to better synthesize functional programs from natural language [39].

**Bayesian Program Learning:** Our work is an instance of Bayesian Program Learning (BPL; see [2, 16, 40, 34, 41]). Previous BPL systems have largely assumed a fixed DSL (but see [34]), and our contribution here is a general way of doing BPL with less hand-engineering of the DSL.

## 7   Discussion

We contribute an algorithm, $EC^2$, that learns to program by bootstrapping a DSL with new domain-specific primitives that the algorithm itself discovers, together with a neural recognition model that learns how to efficiently deploy the DSL on new tasks. We believe this integration of top-down symbolic representations and bottom-up neural networks — both of them learned — helps make program induction systems more generally useful for AI.

A feature of our system is that it learns from (and also, critically *needs*) a corpus of training tasks. Is constructing (or curating) corpra of tasks any easier or better than hand-engineering DSLs? In the immediate future, we expect some degree of hand-engineering of DSLs to continue, especially in domains where humans have strong intuitions about the underlying system of domain-specific concepts, like text editing. However, if program induction is to become a standard part of the AI toolkit, then, in the long-term, we need to build agents that autonomously acquire the knowledge needed to navigate a new domain. So, through the lens of program synthesis, $EC^2$ carries the restriction that it requires a high-quality corpus of training tasks; but, for the program-induction approach to AI, this restriction is a feature, not a bug.

Many directions remain open. Two immediate goals are to integrate more sophisticated neural recognition models [13] and program synthesizers [7], which may improve performance in some domains over the generic methods used here: while our focus in this work was learning to quickly write small programs, we believe more sophisticated neural models, coupled with more powerful program search algorithms, could extend our approach to synthesize larger bodies of code. Another direction is to explore DSL meta-learning: can we find a *single* universal primitive set that could effectively bootstrap DSLs for new domains, including the three domains considered, but also many others?

**Acknowledgments**

We are grateful for collaborations with Eyal Dechter, whose EC algorithm directly inspired this work, and for funding from the NSF GRFP, AFOSR award FA9550-16-1-0012, the MIT-IBM Watson AI Lab, the MUSE program (Darpa grant FA8750-14-2-0242), and an AWS ML Research Award. This material is based upon work supported by the Center for Brains, Minds and Machines (CBMM), funded by NSF STC award CCF-1231216.

## Footnotes

[1]SyGuS text editing problems also prespecify the set of allowed string constants for each task. For these experiments, our system did not use this assistance.

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
