[Supplementary Material]

# Supplement to: Library Learning for Neurally-Guided Bayesian Program Induction

**Kevin Ellis**
MIT
ellisk@mit.edu

**Lucas Morales**
MIT
lucasem@mit.edu

**Mathias Sablé-Meyer**
ENS Paris-Saclay
mathsm@mit.edu

**Armando Solar-Lezama**
MIT
asolar@csail.mit.edu

**Joshua B. Tenenbaum**
MIT
jbt@mit.edu

## 1 An Illustration of the three steps of our algorithm

Below we diagram the steps employed by our algorithm. At each step of the algorithm, we have shaded the observed variables in gray and left the unobserved variables white. Black lines correspond to a connection from the top-down generative model, while red lines correspond to connections from the bottom-up recognition model.

Explore: Infer $p$

Compile: Train $q$
training data: cyan $(x, p)$

Compress: Induce $(\mathcal{D}, \theta)$

## 2 Program Representation

We choose to represent programs using $\lambda$-calculus Pierce (2002). A $\lambda$-calculus expression is either:

- A *primitive*, like the number 5 or the function sum.

- A *variable*, like $x$, $y$, or $z$.

- A *λ-abstraction*, which creates a new function. λ-abstractions have a variable and a body. The body is a λ-calculus expression. Abstractions are written as λvar.body or in Lisp syntax as `(lambda (var) body)`.

- An *application* of a function to an argument. Both the function and the argument are λ-calculus expressions. The application of the function $f$ to the argument $x$ is written as $f\ x$ or as $(f\ \text{x})$.

For example, the function which squares the logarithm of a number is $\lambda x.(\text{square (log } x))$, and the identity function $f(x) = x$ is $\lambda x.x$. The λ-calculus serves as a spartan but expressive Turing complete program representation, and distills the essential features of functional programming languages like Lisp.

However, many λ-calculus expressions correspond to ill-typed programs, such as the program that takes the logarithm of the Boolean `true` (i.e., `log true`) or which applies the number five to the identity function (i.e., $5\ (\lambda x.x)$). We use a well-established typing system for λ-calculus called *Hindley-Milner typing* Pierce (2002), which is used in programming languages like OCaml. The purpose of the typing system is to ensure that our programs never call a function with a type it is not expecting (like trying to take the logarithm of `true`). Hindley-Milner has two important features: Feature 1: It supports *parametric polymorphism*, meaning that types can have variables in them, called *type variables*. Lowercase Greek letters are conventionally used for type variables. For example, the type of the identity function is $\alpha \rightarrow \alpha$, meaning it takes something of type $\alpha$ and return something of type $\alpha$. A function that returns the first element of a list has the type $[\alpha] \rightarrow \alpha$. Type variables are not the same as variables introduced by λ-abstractions. Feature 2: Remarkably, there is a simple algorithm for automatically inferring the polymorphic Hindley-Milner type of a λ-calculus expression Damas & Milner (1982). Our generative model over programs performs Hindley-Milner type inference during sampling: *Unify* in the generative model uses the machinery of Hindley-Milner to ensure that the generated programs have valid polymorphic types. A satisfactory exposition of Hindley-Milner is beyond the scope of this paper, but Pierce (2002) offers a nice overview of lambda calculus and typing systems like Hindley-Milner.

# 3 Generative model over programs

Alg. 1 is a procedure for drawing samples from the generative model $(\mathcal{D}, \theta)$. In practice, we enumerate programs in order of their probability under Alg. 1 rather than sample them.

---

**Algorithm 1** Generative model over programs

---

**function** sample$(\mathcal{D}, \theta, \mathcal{E}, \tau)$:
**Input:** DSL $(\mathcal{D}, \theta)$, environment $\mathcal{E}$, type $\tau$
**Output:** a program whose type unifies with $\tau$
**if** $\tau = \alpha \to \beta$ **then**
    var $\leftarrow$ an unused variable name
    body $\sim$ sample$(\mathcal{D}, \theta, \{\text{var} : \alpha\} \cup \mathcal{E}, \beta)$
    **return** (`lambda` (`var`) body)
**end if**
primitives $\leftarrow \{p | p : \tau' \in \mathcal{D} \cup \mathcal{E}$
                 if $\tau$ can unify with yield$(\tau')\}$
Draw $e \sim$ primitives, w.p. $\propto \theta_e$ if $e \in \mathcal{D}$
                       w.p. $\propto \frac{\theta_{var}}{|\text{variables}|}$ if $e \in \mathcal{E}$
Unify $\tau$ with yield$(\tau')$.
$\{\alpha_k\}_{k=1}^K \leftarrow$ args$(\tau')$
**for** $k = 1$ **to** $K$ **do**
    $a_k \sim$ sample$(\mathcal{D}, \theta, \mathcal{E}, \alpha_k)$
**end for**
**return** $(e\ a_1\ a_2\ \cdots\ a_K)$
**where:**
$$\text{yield}(\tau) = \begin{cases} \text{yield}(\beta) & \text{if } \tau = \alpha \to \beta \\ \tau & \text{otherwise.} \end{cases}$$
$$\text{args}(\tau) = \begin{cases} [\alpha] + \text{args}(\beta) & \text{if } \tau = \alpha \to \beta \\ [] & \text{otherwise.} \end{cases}$$

---

# 4 Neural Recognition Model Architecture

The neural recognition model regresses from an observation (set of input/output pairs: $\{(i_n, o_n)\}_{n \leq N}$) to a $|\mathcal{D}| + 1$ dimensional vector. Each input/output pair is processed by an identical encoder network; the outputs of the encoders are average and passed to an MLP with 1 hidden layer, 32 hidden units, and a ReLU activation:

$$q(x) = \text{MLP}\left(\text{Average}\left(\{\text{encoder}\,(i_n, o_n)\}_{n \leq N}\right)\right) \tag{1}$$

For the string editing and list domains, the inputs and outputs are sequences. Our encoder for these domains is a bidirectional GRU with 64 hidden units that reads each input/output pair; we concatenate the input and output along with a special delimiter symbol between them. We MaxPool the final hidden unit activations in the GRU along both passes of the bidirectional GRU.

For symbolic regression, the input/outputs are densely sampled points along the curve of the function. We rendered these points to a graph, and pass the image of the graph to a convolutional network, which acts as the encoder.

# 5 DSL Induction

## 5.1 Fragment grammars

'Fragment grammars' O'Donnell (2015) is a formalism from computational linguistics developed for the purpose of modeling the reuse of structure in natural language. Here, we will adapt the fragment

$$\text{fragments}(\lambda z.e) = \text{fragments}'(\lambda z.e) \cup \text{fragments}(e)$$
$$\text{fragments}(f\ x) = \text{fragments}'(f\ x) \cup \text{fragments}(f) \cup \text{fragments}(x)$$
$$\text{fragments}(e) = \varnothing, \text{ if } e \text{ is a variable or primitive.}$$

$$\text{fragments}'(\lambda z.e) = \left\{ \lambda z.e' | e' \in \text{fragments}'(e) \right\} \cup \{v\}, v \text{ a free variable}$$
$$\text{fragments}'(f\ x) = \left\{ f'\ x' | f' \in \text{fragments}'(f),\ x' \in \text{fragments}'(x) \right\} \cup \{v\}, v \text{ a free variable}$$
$$\text{fragments}'(e) = \{e\} \cup \{v\}, v \text{ a free variable}, e \text{ a variable or primitive}$$

Figure 1: Procedure for extracting possible fragments from a program. To prevent an explosion in the number of fragments, we also cap the number of free variables in a fragment to 3.

$$\text{closeFragment}(e) = \text{closeFragment}(\lambda z.e), \text{ if } z \text{ is a free variable in } e$$
$$\text{closeFragment}(e) = e, \text{ if } e \text{ has no free variables.}$$

Figure 2: Procedure for converting a fragment to a subroutine.

grammar formalism to model the reuse of structure in a formal language ($\lambda$ calculus), interpreting reuse of structure as subroutines.

In the original fragment grammar formalism, a fragment grammar contains a probabilistic CFG called its *base grammar*; a *fragment* is a tree drawn from the base grammar, but which can contain nonterminal symbols. As a motivating example, consider the following base grammar:

$$S \to (+\ S\ S)$$
$$S \to (\times\ S\ S)$$
$$S \to 0$$
$$S \to 1$$

An example expression drawn from this grammar is $(+\ 1\ (\times\ 0\ 0))$. An example *fragment* drawn from this grammar $(+\ 1\ (\times\ S\ S))$.

When extending fragment grammars to $\lambda$-calculus we write nonterminal symbols as free variables, and use the (current) DSL as our base grammar. For example, if our DSL includes addition, multiplication, and the numbers zero & one (as in our example base grammar), then a possible $\lambda$-fragment would be $(+\ 1\ (\times\ x\ y))$.

In order to use fragments for DSL induction, we need several pieces: (1) a procedure for proposing fragments from programs found in the frontiers (described in figure 1); (2) a procedure for evaluating the likelihood of a program given that a fragment has been added to the DSL (defined in figure 3); and (3) a procedure for converting a fragment to a subroutine (i.e., a closed, typed $\lambda$-calculus expression; described in Figure 2). Putting these procedures together, Algorithm 3 is used to induce a new DSL.

$$\mathbb{P}[p|\mathcal{D},\theta] = P_{\mathcal{D},\theta}(p|\tau,\varnothing), \text{ where } p:\tau$$

$$P_{\mathcal{D},\theta}(\lambda z.p|\alpha \to \beta, \mathcal{E}) = P_{\mathcal{D},\theta}(p|\beta, [\tau \mapsto \alpha]\mathcal{E}), \text{ where } \tau \text{ a fresh type variable}$$

$$P_{\mathcal{D},\theta}(e|\tau,\mathcal{E}) = \underbrace{\sum_{f,xs\in\mathrm{parse}(e)} P'_{\mathcal{D},\theta}(f,xs|\tau,\mathcal{E})}_{\text{Marginalizing out how } \mathcal{D} \text{ produced } e}, \text{ when } e \text{ not a } \lambda\text{-abstraction}$$

$$P'_{\mathcal{D},\theta}(f,xs|\tau,\mathcal{E}) = \underbrace{\sum_{f':\tau_1\to\tau_2\to\cdots\to\tau\in\mathcal{D}} \frac{\theta_f}{Z_{\tau,\mathcal{D},\theta,\mathcal{E}}} \mathbb{1}[\mathrm{match}(f',f) \neq \bot]}_{\text{Marginalize over which fragment produced } f,\, xs} \times$$

$$\prod_{n=1}^{|xs|} P_{\mathcal{D},\theta}(x_n|\tau_n,\mathcal{E}) \prod_{e,\tau'\in\mathrm{match}(f',f)} P_{\mathcal{D},\theta}(e|\tau',\mathcal{E})$$

$$\mathrm{match}(\lambda z.b, \lambda z.b') = \mathrm{match}(b,b')$$

$$\mathrm{match}(f\ x, f'\ x') = \mathrm{match}(f,f') \cup \mathrm{match}(x,x')$$

$$\mathrm{match}(p,p) = \varnothing, p \text{ a primitive or variable bound in fragment}$$

$$\mathrm{match}(v,p) = \{v \mapsto p\}, v \text{ a variable free in fragment}, p \text{ not containing free variables bound in fragment}$$

$$\mathrm{match}(e,e') = \bot, \text{ otherwise.}$$

$$Z_{\tau,\mathcal{D},\theta,\mathcal{E}} = \underbrace{\sum_{f:\tau_1\to\tau_2\to\cdots\to\tau\in\mathcal{D}\cup\mathcal{E}} \theta_f}_{\text{Normalizing constant for type } \tau}$$

$$\mathrm{parse}(f\ x) = \{f', xs + [x] \mid f', xs \in \mathrm{parse}(f)\} \cup \{f\ x\}$$

$$\mathrm{parse}(e) = \{e, []\}, e \text{ not an application}$$

Figure 3: Procedure for calculating likelihood of a program after a new fragment has been added to the DSL. For simplicity we omit typing context and reuse type variables to indicate unification, as is commonly done in e.g. Prolog.

---

**Algorithm 3** DSL Induction Algorithm

---

**Input:** Set of frontiers $\{\mathcal{F}_x\}$
**Hyperparameters:** Pseudocounts $\alpha$, regularization parameter $\lambda$
**Output:** DSL $\mathcal{D}$, weight vector $\theta$
Define $L(\mathcal{D},\theta) = \prod_x \sum_{p\in\mathcal{F}_x} \mathbb{P}[p|\mathcal{D},\theta]$
Define $\theta^*(\mathcal{D}) = \arg\max_\theta \mathrm{Dir}(\theta|\alpha)L(\mathcal{D},\theta)$
Define $\mathrm{score}(\mathcal{D}) = \log \mathbb{P}[\mathcal{D}] + L(\mathcal{D},\theta^*) - \|\theta\|_0$
$\mathcal{D} \leftarrow$ every primitive in $\{\mathcal{F}_x\}$
**while** true **do**
    $N \leftarrow \{\mathcal{D} \cup \{s\}|x \in X, p \in \mathcal{F}_x, s \in \mathrm{fragments}(p)\}$
    $\mathcal{D}' \leftarrow \arg\max_{\mathcal{D}'\in N} \mathrm{score}(\mathcal{D}')$
    **if** $\mathrm{score}(\mathcal{D}') < \mathrm{score}(\mathcal{D})$ **return** $\{\mathrm{closeFragment}(s)|s \in \mathcal{D}\}, \theta^*(\mathcal{D})$
    $\mathcal{D} \leftarrow \mathcal{D}'$
**end while**

---

## 5.2 Estimating $\theta$

We use an EM algorithm to estimate the continuous parameters of the DSL, e.g. $\theta$. Suppressing dependencies on $\mathcal{D}$, the EM updates are

$$\theta = \arg\max_{\theta} \log P(\theta) + \sum_x \mathbb{E}_{Q_x}\left[\log \mathbb{P}\left[p|\theta\right]\right] \qquad (2)$$

$$Q_x(p) \propto \mathbb{P}[x|p]\mathbb{P}[p|\theta] \qquad (3)$$

In the M step of EM we will update $\theta$ by instead maximizing a lower bound on $\log \mathbb{P}[p|\theta]$, making our approach an instance of Generalized EM.

We write $c(e,p)$ to mean the number of times that primitive $e$ was used in program $p$; $c(p) = \sum_{e \in \mathcal{D}} c(e,p)$ to mean the total number of primitives used in program $p$; $R(p)$ to mean the sequence of types input to sample in Alg. 1 of the main paper. Jensen's inequality gives a lower bound on the likelihood:

$$\sum_x \mathbb{E}_{Q_x}\left[\log \mathbb{P}[p|\theta]\right] =$$

$$\sum_{e \in \mathcal{D}} \log \theta_e \sum_x \mathbb{E}\left[c(e,p_x)\right] - \sum_\tau \mathbb{E}\left[\sum_x c(\tau,p_x)\right] \log \sum_{\substack{e:\tau' \in \mathcal{D} \\ \text{unify}(\tau,\tau')}} \theta_e$$

$$= \sum_e C(e) \log \theta_e - \beta \sum_\tau \frac{\mathbb{E}\left[\sum_x c(\tau,p_x)\right]}{\beta} \log \sum_{\substack{e:\tau' \in \mathcal{D} \\ \text{unify}(\tau,\tau')}} \theta_e$$

$$\geq \sum_e C(e) \log \theta_e - \beta \log \sum_\tau \frac{\mathbb{E}\left[\sum_x c(\tau,p_x)\right]}{\beta} \sum_{\substack{e:\tau' \in \mathcal{D} \\ \text{unify}(\tau,\tau')}} \theta_e$$

$$= \sum_e C(e) \log \theta_e - \beta \log \sum_\tau \frac{R(\tau)}{\beta} \sum_{\substack{e:\tau' \in \mathcal{D} \\ \text{unify}(\tau,\tau')}} \theta_e$$

where we have defined

$$C(e) \triangleq \sum_x \mathbb{E}\left[c(e,p_x)\right]$$

$$R(\tau) \triangleq \mathbb{E}\left[\sum_x c(\tau,p_x)\right]$$

$$\beta \triangleq \sum_\tau \mathbb{E}\left[\sum_x c(\tau,p_x)\right]$$

Crucially it was defining $\beta$ that let us use Jensen's inequality. Recalling from the main paper that $P(\theta) \triangleq \text{Dir}(\alpha)$, we have the following lower bound on M-step objective:

$$\sum_e (C(e) + \alpha) \log \theta_e - \beta \log \sum_\tau \frac{R(\tau)}{\beta} \sum_{\substack{e:\tau' \in \mathcal{D} \\ \text{unify}(\tau,\tau')}} \theta_e \qquad (4)$$

Differentiate with respect to $\theta_e$, where $e : \tau$, and set to zero to obtain:

$$\frac{C(e) + \alpha}{\theta_e} \propto \sum_{\tau'} \mathbb{1}\left[\text{unify}(\tau,\tau')\right] R(\tau') \qquad (5)$$

$$\theta_e \propto \frac{C(e) + \alpha}{\sum_{\tau'} \mathbb{1}\left[\text{unify}(\tau,\tau')\right] R(\tau')} \qquad (6)$$

The above is our estimator for $\theta_e$. Despite the convoluted derivation, the above estimator has an intuitive interpretation. The quantity $C(e)$ is the expected number of times that we used $e$. The

quantity $\sum_{\tau'} \mathbb{1}\left[\text{unify}(\tau, \tau')\right] R(\tau')$ is the expected number of times that we *could have* used $e$. The hyperparameter $\alpha$ acts as pseudocounts that are added to the number of times that we used each primitive, and are not added to the number of times that we could have used each primitive.

We are only maximizing a lower bound on the log posterior; when is this lower bound tight? This lower bound is tight whenever all of the types of the expressions in the DSL are not polymorphic, in which case our DSL is equivalent to a PCFG and this estimator is equivalent to the inside/outside algorithm. Polymorphism introduces context-sensitivity to the DSL, and exactly maximizing the likelihood with respect to $\theta$ becomes intractable, so for domains with polymorphic types we use this estimator.

# 6 Hyperparameters & Implementation Details

We set structure penalty $\lambda = 1$ (Eq. 5 of the main paper) and smoothness parameter $\alpha = 10$ (Eq. 6 of the main paper) for all experiments. For list processing and text editing we used a search timeout of two hours; because the symbolic regression problems are easier, we used a timeout of only five minutes for these.

Because the frontiers can become very large in later iterations of the algorithm, we only keep around the top $10^4$ programs in the frontier $\mathcal{F}_x$ as measured by $\mathbb{P}[x, p | \mathcal{D}, \theta]$.

# 7 Why not the ELBO Bound?

Our lower bound $\mathscr{L}$ is unconventional, and one might wonder why we do not instead maximize an ELBO-style bound like in a VAE or in the EM algorithm. Surprisingly, maximizing an ELBO-style bound leads to a pathological behavior that causes the model to easily become trapped in local optima.

If we were to maximize the ELBO bound to perform inference in our generative model, then, during DSL induction, we would seek a new $(\mathcal{D}^*, \theta^*)$ maximizing the following lower bound on the likelihood (along with an unimportant regularizing term on the DSL):

$$\sum_{x \in X} \mathbb{E}_{p \sim Q_x}\left[\log \mathbb{P}[p | \mathcal{D}^*, \theta^*]\right] \tag{7}$$

$$Q_x(p) \triangleq \mathbb{P}[p | x, \mathcal{D}, \theta] \tag{8}$$

where $(\mathcal{D}, \theta)$ is our current estimate of the generative model. These equations fall out of an EM-style derivation, and one could replace $Q_x(p)$ with the recognition model $q(p|x)$, either using importance sampling (so the expectation in Eq. 7 is taken over $q$ and we reweigh using $Q_x$) or by directly using $q$ as our approximate posterior over the program that solves task $x$.

We do not maximize a bound of this form because it takes an expectation over the *previous* iteration's posterior over programs, so the approximate posterior $Q_x$ at the next iteration ends up being very close to previous approximate posterior. Intuitively, we want the DSL induction to be a function *only* of the programs that we have found, and *not* be a function of how the previous generative model weighed them. In practice, we found that maximizing EM-style bounds, like the ELBO, leads to a kind of hysteresis effect, where the next generative model too closely matches the previous one, causing the algorithm to easily become trapped in local optima.

# 8 List Processing Data Set

Each list processing tasks we created is in described in Tbl 1.

# 9 Learned DSLs

Here we present representative DSLs learned by our model. DSL primitives discovered by the algorithm are prefixed with #. Variables are prefixed with $, and we adopt De Bruijn indices to model bound variables Pierce (2002).

| | |
|---|---|
| add-k for k ∈ {0..5} | kth-largest for k ∈ {1..5} |
| append-index-k for k ∈ {1..5} | kth-smallest for k ∈ {1..5} |
| append-k for k ∈ {0..5} | last |
| bool-identify-geq-k for k ∈ {0..5} | len |
| bool-identify-is-mod-k for k ∈ {1..5} | max |
| bool-identify-is-prime | min |
| bool-identify-k for k ∈ {0..5} | modulo-k for k ∈ {1..5} |
| caesar-cipher-k-modulo-n | mult-k for k ∈ {0..5} |
|     for k ∈ {0..5} and n ∈ {1..5} | odds |
| count-head-in-tail | pop |
| count-k for k ∈ {0..5} | pow-k for k ∈ {1..5} |
| drop-k for k ∈ {0..5} | prepend-index-k for k ∈ {1..5} |
| dup | prepend-k for k ∈ {0..5} |
| empty | product |
| evens | range |
| fibonacci | remove-empty-lists |
| has-head-in-tail | remove-eq-k for k ∈ {0..3} |
| has-k for k ∈ {0..5} | remove-gt-k for k ∈ {0..3} |
| head | remove-index-k for k ∈ {1..5} |
| index-head | remove-mod-head |
| index-k for k ∈ {1..5} | remove-mod-k for k ∈ {2..5} |
| is-evens | repeat |
| is-mod-k for k ∈ {1..5} | repeat-k for k ∈ {1..5} |
| is-odds | repeat-many |
| is-primes | replace-all-with-index-k for k ∈ {1..5} |
| is-squares | reverse |
| keep-eq-k for k ∈ {0..3} | rotate-k for k ∈ {1..5} |
| keep-gt-k for k ∈ {0..3} | slice-k-n for k ∈ {1..5} and n ∈ {1..5} |
| keep-mod-head | sort |
| keep-mod-k for k ∈ {1..5} | sum |
| keep-primes | tail |
| keep-squares | take-k for k ∈ {1..5} |

Table 1: Our list processing data set

## 9.1 List processing

```
#(+ 1 1)
#(λ (cdr (cdr $0)))
#(λ (foldr $0 1 (λ (λ (∗ $0 $1)))))
#(λ (cons (car $0) nil))
#(λ (λ (foldr $0 $1 (λ (λ (cons $1 $0))))))
#(λ (λ (foldr $0 (is−nil $0) (λ (λ (if $0 $0 (eq? $3 $1)))))))
#(λ (map (λ (eq? $1 $0))))
#(λ (∗ $0 (∗ $0 $0)))
#(+ 1 #(+ 1 1))
#(λ (map (λ (gt? $0 $1))))
#(λ (foldr $0 nil (λ (λ (if (is−square $1) (cons $1 $0) $0)))))
#(λ (map (λ (eq? $0 (length (range $0)))) $0))
#(λ (λ (map (λ (index $0 $1)) (range $1))))
#(λ (cdr (#(λ (cdr (cdr $0))) $0)))
#(λ (map (λ (index 1 $1))))
#(λ (foldr $0 nil (λ (λ (cons $1 (cons $1 $0))))))
#(λ (λ (cons (car $0) $1)))
#(+ 1 #(+ 1 #(+ 1 1)))
#(λ (map (λ (gt? 1 (mod $0 $1)))))
#(λ (λ (map (λ (mod (+ $0 $1) $2)))))
#(λ (#(λ (λ (foldr $0 $1 (λ (λ (cons $1 $0)))))) (#(λ (λ (foldr $0 $1 (λ
  ↪ (λ (cons $1 $0)))))) $0 $0) $0))
#(λ (λ (#(λ (λ (foldr $0 $1 (λ (λ (cons $1 $0)))))) (cons $0 nil) $1)))
#(+ #(+ 1 #(+ 1 1)) #(+ 1 1))
#(λ (map (λ (+ #(+ 1 #(+ 1 1)) (+ $1 $0)))))
#(λ (map (λ (+ $0 $1))))
#(λ (λ (foldr $0 (is−nil $0) (λ (λ (gt? $1 (#(λ (∗ $0 (∗ $0 $0)))
  ↪ $3)))))))
#(λ (foldr $0 0 (λ (λ (+ $0 (#(λ (foldr $0 1 (λ (λ (∗ $0 $1))))) (range
  ↪ $1)))))))
#(λ (#(λ (cdr (#(λ (cdr (cdr $0))) $0))) (cdr $0)))
#(λ (#(λ (foldr $0 nil (λ (λ (if (is−square $1) (cons $1 $0) $0)))))
  ↪ (map (λ (∗ $0 (+ $0 $0))) $0)))
#(λ (λ (#(λ (λ (foldr $0 $1 (λ (λ (cons $1 $0)))))) (#(λ (cons (car $0)
  ↪ nil)) $0) $1)))
#(λ (λ (length (#(λ (#(λ (foldr $0 nil (λ (λ (if (is−square $1) (cons $1
  ↪ $0) $0))))) (map (λ (∗ $0 (+ $0 $0))) $0))) (map (λ (− $1 $0))
  ↪ $1)))))
#(λ (λ (is−square (#(λ (foldr $0 1 (λ (λ (∗ $0 $1))))) (#(λ (λ (map (λ
  ↪ (mod (+ $0 $1) $2))))) (length (#(λ (#(λ (λ (foldr $0 $1 (λ (λ
  ↪ (cons $1 $0))))))) (#(λ (λ (foldr $0 $1 (λ (λ (cons $1 $0)))))) $0
  ↪ $0)) $0)) $1 $0)))))
#(λ (λ (foldr (cdr $0) $1 (λ (λ (#(λ (λ (#(λ (λ (foldr $0 $1 (λ (λ (cons
  ↪ $1 $0))))))) (#(λ (cons (car $0) nil)) $0) $1))) (cdr $0) $0))))))
#(λ (is−nil (#(λ (#(λ (foldr $0 nil (λ (λ (if (is−square $1) (cons $1
  ↪ $0) $0))))) (map (λ (∗ $0 (+ $0 $0))) $0))) (#(λ (λ (map (λ (mod
  ↪ (+ $0 $1) $2))))) #(+ 1 1) 1 $0))))
#(λ (λ (gt? (#(λ (λ (length (#(λ (#(λ (foldr $0 nil (λ (λ (if (is−square
  ↪ $1) (cons $1 $0) $0))))) (map (λ (∗ $0 (+ $0 $0))) $0))) (map (λ
  ↪ (− $1 $0)) $1))))) $0 $1) 1)))
```

## 9.2 Text editing

```
#(+ 1)
#(λ (λ (fold $0 $0 (λ (λ (if (char−eq? $1 $3) nil (cons $1 $0)))))))
#(λ (λ (fold $0 $0 (λ (λ (cdr (if (char−eq? $1 $3) $2 $0)))))))
#(λ (λ (fold $0 $1 (λ (λ (cons $1 $0))))))
#(λ (λ (#(λ (λ (fold $0 $1 (λ (λ (cons $1 $0)))))) (cons $0 $1))))
#(λ (#(λ (λ (λ (cons (car $0) (cons $1 $2))))) (#(#(λ (λ (λ (cons (car
  ↪ $0) (cons $1 $2))))) nil) '.' $0) '.'))
#(λ (λ (fold $0 $0 (λ (λ (fold $0 $0 (λ (λ (if (char−eq? $1 $5) (cdr $2)
  ↪ $0)))))))))
#(λ (λ (map (λ (if (char−eq? $1 $0) $2 $0)))))
```

```
#(λ (#(λ (λ (fold $0 $1 (λ (λ (cons $1 $0)))))) $0 STRING))
#(λ (map (λ (index $0 $1))))
#(λ (unfold $0 (λ (nil? $0)) (λ (car $0)) (λ (#(λ (λ (fold $0 $0 (λ (λ
    ↪ (cdr (if (char−eq? $1 $3) $2 $0))))))) SPACE $0))))
#(#(λ (λ (λ (cons (car $0) (cons $1 $2))))) nil)
```

### 9.3 Symbolic regression

```
#(λ (/. (/. REAL $0) $0))
#(λ (+. $0 REAL))
#(λ (#(λ (+. $0 REAL)) (∗. $0 (#(λ (#(λ (+. $0 REAL)) (∗. (#(λ (#(λ (#(λ
    ↪ (+. $0 REAL)) (∗. $0 REAL))) (∗. (#(λ (+. $0 REAL)) $0) $0))) $0)
    ↪ $0))) $0))))
#(λ (/. (#(λ (/. (/. REAL $0) $0)) $0) $0))
#(λ (λ (#(/. REAL) (/. (#(λ (+. $0 REAL)) $0) $1))))
#(λ (#(λ (+. $0 REAL)) (#(λ (#(/. REAL) (#(λ (+. $0 REAL)) $0))) $0)))
#(#(λ (λ (#(/. REAL) (/. (#(λ (+. $0 REAL)) $0) $1)))) (#(λ (/. (#(λ (/.
    ↪ (/. REAL $0) $0)) $0) $0)) REAL))
#(λ (/. (#(λ (#(λ (#(λ (+. $0 REAL)) (∗. $0 REAL))) (∗. (#(λ (+. $0
    ↪ REAL)) $0) $0))) $0) (#(λ (+. $0 REAL)) $0)))
```