[Reviews · NeurIPS 2018]

Reviewer 1



= Summary A method for learning a DSL for program synthesis together with a search algorithm in that DSL is presented. The method proceeds iteratively, trying to solve tasks with the current DSL, and then extracting new DSL components from the solutions. Experiments show that bootstrapping the method with a DSL made up of trivial primitives is sufficient to discover common high-level constructs present in manually constructed DSLs. The paper tackles an important problem (DSL design) in an elegant and novel way. The clarity of the paper is not perfect, as the details of the idea require more space than the 8 pages available, but it clearly is stepping stone towards a new generation of program synthesis approaches. = Quality The paper precisely defines its main algorithm and the core concepts of the approach. There are a few omissions (e.g., Sect. 2.4 does not explain the exact mechanism to extract new constructs; the supplement gives an algorithm, but omits a clear definition of how "fragments" are found), but overall the paper is precise and explains core concepts well. The experiments are very thorough, comparing the method to interesting ablations and baselines. What I am missing is an analysis of the influence of the initial set of primitives. For example, the list processing set starts with a non-minimal set of primitives (as length is "foldr (\lambda (n) -> 1 + n) 0" and index can be implemented using the others as well, though painfully). What happens, for example, if length is removed? As an additional data point, I would have appreciated a look at the "evolution" of solutions to a fixed task through the iterations of the main algorithm, i.e., show an initial, early and late solution in the style of Table 1. = Clarity Many part of the paper are better understood with the supplement in hand (Sect. 2.3 is not discussing details of how q is modeled, but Supplement Sect. 5 makes clear that is is essentially like DeepCoder; Sect. 2.4 is missing details as discussed above). Overall, the paper is easily understood, but not entirely self-contained and not precise enough to allow re-implementation by third parties. The supplement is very unpolished (there are many grammatical errors and half-finished sentences, e.g. in line 6, line 33, lines 45/46, ...) = Originality While drawing on the program synthesis literature of the last few years (e.g., the guided enumerative search), the core contributions of the paper are completely new and are a substantial step forward for the field. = Significance Designing a suitable DSL is often one of the hardest tasks in applying program synthesis to a new domain; the promise of automatically learning a DSL (together with a search algorithm) is obvious. I am not entirely convinced by the authors idea of generalizing to other generative programs, but I am looking forward to being proven wrong.

Reviewer 2



This paper proposes an algorithm for program synthesis in an incremental way. There are three steps to it: 1) searching for program solutions, 2) expandind the dsl by discovering useful reusable components, 3) training a neural network that primes the program solution search in 1). The interesting part of the contribution is the focus on roughly mimicking the way humans refactor, recompose and reuse when coding. Strengths - the problem presented is interesting, and the scc algorithm is novel. the theoretical framework is sound, and approximations made are reasonable and useful - the paper presents a good evaluation framework on three different tasks and through the ablation analysis that clearly shows the contributions of each part of the model to the final results - the results are impressive - excellent performance on all three tasks, both accuracy-wise and time-wise Weaknesses - this is quite a dense paper and it packs a lot of information - the paper could benefit from a comparison to a well known systems such as flashfill, where the dsl is fixed Questions: - 125-126 the probability of a variable occurring in a program. could you provide an example here? - equation 3 - is this the counting norm? if so, can you provide a quick rationale why it is used - 167 - can you explicitly say what AIC is and/or cite it? - 181-183 - how is this unification done? - 200 - was this manually created? - what was the recognition model validated on? Other: - figure 4 position is weird - figure 4 bottom left and right are not clear update: reading the author response I'm firmly standing by my score - I want to see this paper accepted. The authors addressed all the questions I had, and all the other reviewers' concerns. The paper will still stay densely packed with info, and I hope the authors will fix the issues of the appendix as otherwise it will be difficult to follow the main body of the paper without it.

Reviewer 3



The authors present an inductive programming approach where a DSL is learned by a search, compile, and compress strategy. They evaluate their approach in three different domains -- list processing, text editing, and symbolic regression. They motivate their approach by pointing out the effectivenes and flexibility of human learning. Similar arguments are presented by Schmid & Kitzelmann, Cognitive Systems Research 2011 and the approach has some similarities with the MagicHaskeler of Katayama.Although some relation to pevious research is missing (mainly the two groups named above), the presented approach is original. I am most impressed by the empirical demonstration of applicability in three rather distinct domains (but: see also Schmid/Kitzelmann, 2011). The presentation is technically sound and the empirical evaluation is very convincing.